# Alternative PCR-Based Approaches for Generation of *Komagataella phaffii* Strains

**DOI:** 10.3390/microorganisms11092297

**Published:** 2023-09-12

**Authors:** Anastasiya Makeeva, Dmitry Muzaev, Maria Shubert, Tatiana Ianshina, Anton Sidorin, Elena Sambuk, Andrey Rumyantsev, Marina Padkina

**Affiliations:** Laboratory of Biochemical Genetics, Department of Genetics and Biotechnology, Saint Petersburg State University (SPBU), Saint Petersburg 199034, Russia

**Keywords:** *Komagataella phaffii*, *Pichia pastoris*, heterologous protein production, split markers, multicopy strains, Neo-2/15

## Abstract

*Komagataella phaffii* (*Pichia pastoris*) is a widely known microbial host for recombinant protein production and an emerging model organism in fundamental research. The development of new materials and techniques on this yeast improves heterologous protein synthesis. One of the most prominent ways to enhance protein production efficiency is to select *K. phaffii* strains with multiple expression cassettes and generate Mut^S^ strains using various vectors. In this study, we demonstrate approaches to expand the applications of pPICZ series vectors. Procedures based on PCR amplification and in vivo cloning allow rapid exchange of selectable markers. The combination of PCR amplification with split-marker-mediated transformation allows the development of *K. phaffii* Mut^S^ strains with two expression cassettes using pPICZ vectors. Both PCR-based approaches were applied to efficiently produce interleukin-2 mimetic Neo-2/15 in *K. phaffii*. The described techniques provide alternative ways to generate and improve *K. phaffii* strains without the need for obtaining new specific vectors or additional cloning of expression cassettes.

## 1. Introduction

*Komagataella phaffii* (*Pichia pastoris*) yeast has been successfully employed in biotechnology as an excellent microbial host for the production of recombinant proteins and organic compounds [1,2,3,4]. This yeast serves as a microbial cell factory for small-scale synthesis of recombinant proteins and is easily adaptable for large-scale industrial production. Researchers now use *K. phaffii* to synthethize various proteins for different applications in medicine [5], plant science [6], molecular biology structural analysis [7] and others [8,9]. The practical importance of *K. phaffii* and its growing recognition among researchers allowed this yeast to become a new model organism in fundamental research [10]. “Omics” technologies (genomics [11], transcriptomics [12,13], proteomics [14], lipidomics [15] and metabolomics [16]) have greatly accelerated the use of *K. phaffii*.

To synthesize a recombinant protein in *K. phaffii* cells, it is necessary to first deliver the expression cassette. Various vectors are available for this purpose. Although episomal plasmids are being developed for *K. phaffii* [17,18], integrative vectors are still used in most studies [3]. Integrative plasmids are usually “shuttle” vectors that can be propagated in *Escherichia coli* and then transformed into *K. phaffii* with integration into the genome. Such vectors consist of (1) the bacterial origin of replication, (2) an expression cassette and (3) markers (or markers) for the selection of transformants. Here, pPIC9 and pPICZ series vectors (Invitrogen, now Thermo Fisher Scientific, Waltham, MA USA) can serve as representative examples (Figure 1).

pPIC9 and pPICZ series vectors contain bacterial origins for plasmid propagation in *E. coli* (pBR322 and pUC, respectively), and an expression cassette comprising *AOX1* promoter, multi-cloning site and *AOX1* terminator sequences. pPIC9 and pPICZalpha vectors contain a sequence-encoding alphaMF N-terminal signal from *Saccharomyces cerevisiae* to secrete the synthesized protein into the media. pPICZ series vectors also contain c-myc epitope and 6xhis-tag sequences which can be added to the recombinant protein.

A large number of different promoters can be used for heterologous expression in *K. phaffii* cells [3,19]. However, due to its excellent properties, the alcohol oxidase *AOX1* gene promoter is still the most popular choice. It provides very high transcription levels (up to 5% of total mRNA) and tight carbon source regulation [20,21].

As for selectable markers in yeast transformation, pPIC9 and pPICZ series vectors enable auxotrophic and antibiotic selection, respectively. pPIC9 uses the *K. phaffii HIS4* gene as an auxotrophic marker, encoding a trifunctional enzyme (histidinol dehydrogenase, phosphoribosyl-ATP-cyclohydrase and phosphoribosyl-ATP-pyrophosphohydratase) [22]. Therefore, and due to its importance in histidine biosynthesis, the *HIS4* gene serves as a selectable marker, allowing to transform *K. phaffii* strains carrying the corresponding deletion (e.g., GS115 (*his4*)) [23,24]. Other selectable markers (e.g., *ARG4*, *ADE1* and *URA3*) and related auxotrophic strains are also available for *K. phaffii* [25,26]. The pPIC9 vector also contains the beta-lactamase gene, conferring ampicillin resistance as a separate marker for selection in bacteria.

The pPICZ series vectors carry the *Sh ble* gene from *Streptoalloteichus hindustanus* conferring Zeocin™ resistance. Since bacteria and yeast are susceptible to this antibiotic, the *Sh ble* gene serves as a single selectable marker for both microorganisms. To provide gene expression in different hosts, both the synthetic prokaryotic promoter *EM7* and the promoter from the *S. cerevisiae TEF1* gene are placed adjacently upstream (5′) of the recombinant protein coding sequence. Other antibiotic-based selectable markers providing resistance to G418, blasticidin, hygromycin B and nourseothricin are also efficiently used during *K. phaffii* transformation [27,28,29].

Selectable markers have different advantages and limitations. For example, auxotrophic markers allow for the selection of yeast transformants on simple minimal media, but they require the corresponding auxotrophic strains. The growth of such strains may be impaired by mutations in the corresponding genes [25] and their cultivation requires additional cell culture media supplements [30]. In contrast, methods based on antibiotic resistance markers can be used to transform wild-type strains, but they require the addition of antibiotics to the growth media. 

Auxotrophic markers are used for yeast transformation, i.e., different markers are additionally required for bacterial transformation. In contrast, antibiotic resistance markers can be organized in a single gene expressed in both bacterial and yeast cells, allowing the vectors to be smaller in size in comparison to their counterparts with auxotrophic markers (3597 bp for pPICZalpha B vs. 8023 bp for pPIC9).

After small-scale pilot experiments, there are multiple ways to improve the efficiency of recombinant protein production in *K. phaffii* for higher levels of protein yield. For example, it is possible to generate a strain with multiple expression cassettes [31,32,33]. This often includes the use of multiple selectable markers or dosage sensitive markers [32,34,35].

The use of multiple selectable markers is hampered by the need to clone the gene of interest in several plasmids. To avoid this, in the first part of this study, we provide an approach for the rapid generation of *K. phaffii* vectors with different selectable markers. This approach is based on PCR amplification of parts of the pPICZ series vector followed by in vivo cloning (iVEC) of *E. coli*.

The use of dosage sensitive markers for selection of *K. phaffii* strains with multiple expression cassettes can be demonstrated using pPICZ series vectors. These vectors carry antibiotic resistance markers and directly allow the selection of multiple integration events [35] and amplification of expression cassettes after transformation [36]. These methods are based on the use of high concentrations of antibiotics that have to be compensated in *K. phaffii* cells by multiple copies of the integrated vector. Multicopy integration can also be mediated by using defective auxotrophic markers [34] or by combining auxotrophic and antibiotic resistance markers in one vector (e.g., pPIC9K).

Another way to improve the production of recombinant protein is the generation of Mut^S^ strains (methanol utilization slow) [37,38]. Such strains carry a deletion of the *AOX1* gene, resulting in impaired methanol consumption. This may lead to an increase in the production of recombinant protein because the cellular energy economy is not directed toward the synthesis of high amounts of alcohol oxidase. Also, methanol, which is needed for *AOX1* promoter induction and heterologous gene expression, is now consumed by Mut^S^ strains at lower rates.

The pPIC9 vector allows us to directly generate Mut^S^ strains through two acts of recombination between the vector and *AOX1* locus in the *K. phaffii* genome, replacing the *AOX1* locus with the expression cassette and *HIS4* gene [39]. The *AOX1* promoter serves as a homology arm for such recombination. As for the second homology arm, pPIC9 additionally contains 3′ sequence from the *AOX1* locus.

pPICZ series vectors do not provide a possibility to directly generate *K. phaffii* Mut^S^ strains in the absence of *AOX1* deletion as a prerequisite. To bypass this restriction, in the second part of this study we provide a new approach to generate Mut^S^ strains with only vectors that are based on the pPICZ series. This approach is also based on PCR amplification of parts of the vector, followed by *K. phaffii* transformation using the Zeocin™ resistance gene as a split marker. 

Different materials, methods and *K. phaffii* strains have varying levels of availability in labs around the world. Given the long-term market presence, pPICZ vectors have been widely used for heterologous protein production in *K. phaffii* [3,35,40]. On the practical side, many gene synthesis companies offer pPICZ vectors to clone synthesized DNA fragments, and most *K. phaffii* vectors deposited in the Addgene repository have a pPICZ series plasmid as the backbone. Therefore, many researchers globally are familiar with the vector. In this work, we introduce PCR-based approaches to provide researchers with alternative means to generate productive *K. phaffii* strains using existing pPICZ vectors.

## 2. Materials and Methods

### 2.1. Bacterial and Yeast Strains

The bacterial strain *E. coli* DH5α (*F′phi80dlacZ delta (lacZYA_argF) U169 deoRrecA1 endA1 hsdR17(rK–mK+) phoA supE44lambda_thi_1 gyrA96 relA1/F′ proAB+ lacIqdeltaM15 Tn10(tetr)*) (Thermo Fisher Scientific, Waltham, MA, USA) was used for the construction of plasmids and in vivo cloning (iVEC).

*K. phaffii* GS115 (*his4*) and X-33 strains (Thermo Fisher Scientific, Waltham, MA USA) were used as parental yeast strains. The developed *K. phaffii* yeast strains are listed in Table 1.

### 2.2. Media and Cultivation Conditions

Standard LB medium was used for manipulations with *E. coli*. In total, 1L of LB contained (here and further *w*/*v*): 1% tryptone, 0.5% yeast extract, 1% NaCl, and 2.4% agar for plates. TYM medium was used to prepare *E. coli*-competent cells for transformation. In total, 1L of TYM contained 2% trypton, 0.5% yeast extract, 100 mM NaCl and 10 mM MgSO_4_.

YPD media was used for routine manipulations with *K. phaffii* strains. In total, 1L of YPD contained 2% glucose, 2% peptone, 1% yeast extract, and 2.4% agar. YEPDS media was used for Zeocin™ and G418 selection. In total, 1L of YEPDS contained 2% glucose, 2% peptone, 1% yeast extract, 18.2% sorbitol (1 M), 2.4% agar, and 200 mg Zeocin™ or 400 mg G418. Minimal Dextrose medium (MD) was used for selection of His^+^ transformants. In total, 1L of MD medium contained 1.34% YNB, 4 × 10^−5^% biotin, 2% dextrose, and 2.4% agar. Minimal Methanol medium (MM) was used for selection of Mut^S^ transformants and for qualitative analysis of reporter gene activity. In total, 1L of MM contained 1.34% YNB, 4 × 10^−5^% biotin, 0.5% methanol, and 2.4% agar.

Buffered Glycerol-complex Medium (BMGY) and Buffered Methanol-complex Medium (BMMY) were used for quantitative analysis of reporter acid phosphatase activity and to study Neo-2/15 recombinant protein production. In total, 1L of BMGY and BMMY contained 1% yeast extract, 2% peptone, 100 mM potassium phosphate (pH 6.0), 1.34% YNB, 4 × 10^−5^% biotin and 1% glycerol or 0.5% methanol, respectively.

Synthesis of recombinant proteins Neo-2/15 and Pho5 for quantitative analysis was performed in two steps. First, *K. phaffii* cells were inoculated into 20 mL of BMGY in three replicates and cultivated for 46 h. Second, cells from each sample were collected by centrifugation (10 min at 5000 rpm, 3000× *g*), transferred into 20 mL of BMMY and cultivated for 48 h. After 24 h of cultivation in BMMY, cultures were supplemented with an additional 0.5 mL of methanol. 

### 2.3. Molecular Methods 

Plasmid extraction was carried out using the “Plasmid Miniprep kit” (Evrogen, Moscow, Russia). DNA fragments were purified from reaction mixes and agarose gels using the “Cleanup Standard kit” (Evrogen, Moscow, Russia). Genomic DNA was isolated from yeast cells using “LumiPure from AnySample kit” (Lumiprobe, Moscow, Russia). DNA concentration was measured using a Nanodrop™ 2000c spectrophotometer (Thermo Fisher Scientific, Waltham, MA USA).

Enzymatic reactions were performed using the buffers and conditions recommended by the manufacturers. *XhoI* (*Sfr274I*), *XbaI*, *SacI* (*Psp124B I*) and *AgeI* (*AsiGI*) (SybEnzyme, Novosibirsk, Russia) restriction endonucleases were used for DNA hydrolysis. Dephosphorylation of vectors was performed using FastAP thermosensitive alkaline phosphatase (Thermo Fisher Scientific, Waltham, MA USA). DNA ligation was carried out using T4 ligase (Evrogen, Moscow, Russia).

### 2.4. Electrophoresis Methods

DNA electrophoresis in agarose gel was performed according to [41].

Precipitation of proteins from a yeast culture medium was conducted by the ammonium sulfate precipitation method [42]. Cultures were centrifuged at 7000 rpm for 5 min (3300× *g*); the supernatant was used for next steps of protein purification. Fine (NH_4_)_2_SO_4_ powder was added to 1 mL of medium with thorough mixing until 60% saturation. The solution was incubated for 30 min at +4 °C and then centrifuged for 15 min at 13,400 rpm (12,100× *g*). The resulting precipitate was dissolved in 50 µL of 10 mM Tris buffer and used for further analysis.

Protein SDS-PAGE electrophoresis and Coomassie blue G-250 gel staining were performed according to [43]. Standard 6–15% gels and Novex™ Tris-Glycine gels (gradient 8–16%) were used. To quantify gel band density and estimate the amount of protein in samples, gel images were analyzed using ImageJ [44]. The density of each lane was analyzed three times, and the average density value for each sample was calculated. Based on the resulting values, the average density for three replicates of each strain was calculated. The obtained values were used for the calculation of the difference in the amount of synthesized protein (Appendix A).

### 2.5. Western Blot Hybridization

Following protein separation by SDS-PAGE, the gel was incubated in renaturation buffer (0.1 mM DTT, 10 mM Tris (pH 7.5), 20 mM EDTA, 50 mM glycine, 4 M urea) for 10 min, and then for 10 min in transfer buffer (25 mM Tris, 192 mM glycine, 20% methanol). Then, the gel was placed in a “transfer sandwich” (pad–filter paper–gel–nitrocellulose membrane–filter paper–pad) in a transfer buffer. The transfer was performed for 1 h at 70 V. Next, the membrane was incubated in a TBS-Casein buffer (Tris-Buffered Saline with 1% (*w*/*v*) Casein buffer, Bio-Rad Laboratories, Hercules, CA USA) overnight at room temperature. The membrane was transferred into 5 mL of TBS-Casein buffer with 1μg/mL of primary antibodies specific to the c-myc epitope (PSM103-100, SCI-STORE, Moscow, Russia) and incubated for 1.5 h at room temperature. After incubation, the membrane was washed with TBS-Casein buffer for 5 min three times and incubated in 5 mL of TBS-Casein buffer with rabbit anti-mouse antibodies conjugated with horseradish peroxidase (AP160P, SigmaAldrich, Burlington, MA USA) for 1 h. Finally, the membrane was washed in 10 mM tris-HCl (pH 8.0) for 5 min three times. For membrane staining, the reaction of 3,3′-diaminobenzidine with 3% hydrogen peroxide was performed in 10 mL of 10 mM tris-HCl.

### 2.6. PCR Amplification

Primers for PCR are listed in Table 2.

Analytical PCR was performed using “Encyclo Polymerase kit” (Evrogen, Moscow, Russia). A “Thersus Polymerase kit” (Evrogen, Moscow, Russia) was used when DNA fragments were amplified for further cloning into vectors, for the iVEC procedure, or for yeast transformation. Gradient PCR was performed for each fragment to determine the optimal primer annealing temperature (in the range from 40 to 60 °C). In further experiments, a 53 °C annealing temperature was used for all primers. The general PCR program was as follows: (1) 95 °C—3 min, (2) 30 cycles: 95 °C—30 s, 53 °C—30 s, 72 °C—1 min per kb, (3) 72 °C—2 min.

Real-time PCR experiments were performed using the “Reagent kit for real-time PCR in the presence of EVA Green dye” (Syntol, Moscow, Russia) and the “CFX96 Touch Real-Time PCR Detection System” (Bio-Rad Laboratories, Hercules, CA USA). The program was as follows: (1) 95 °C—300 s, (2) 45 cycles: 95 °C—30 s, 58 °C—30 s, 72 °C—30 s. For each analyzed strain, four technical replicates were performed to amplify both the Neo-2/15 gene using NeoRQ-F and NeoRQ-R primers and the *K. phaffii* reference actine gene (PAS_chr3_1169) using ACT1-F and ACT1-R primers. Results were analyzed using the 2(-Delta Delta C(T)) method [45].

In PCR reactions, 20 ng of plasmid DNA or 100 ng of yeast genomic DNA was used as a template. For PCR screening of *E. coli* transformants (colony PCR), single colonies were picked using sterile pipette tips and re-suspended in 10 µL of water. In total, 1 µL of each probe was used for the PCR reaction. After analysis of PCR results using agarose gel electrophoresis, the remaining 9 µL of *E. coli* suspension was spread on antibiotic-containing LB plates. 

### 2.7. Vector Construction

pPIC9-PHO5 and pPIC9-eGFP vectors were generated previously [46,47]. The pPIC9-PHO5 vector contains the coding sequence of the *S. cerevisiae PHO5* acid phosphatase gene (YBR093C) that was cloned into the pPIC9 plasmid using *BamHI* and *EcoRI* restriction sites. To construct pPIC9-eGFP, the eGFP sequence was first amplified from pCMV-GFP [48] and then cloned into pPIC9 plasmid using *BamHI* and *EcoRI* restriction sites. For pPICZ-PHO5 and pPICZ-eGFP plasmids, the insert fragments were cut out of pPIC9-PHO5 and pPIC9-eGFP using *SacI* and *AgeI* restriction enzymes. pPICZalpha B was cut using *SacI* and *AgeI* restriction enzymes and dephosphorylated using the FastAP enzyme. Vector and insert fragments were separated using agarose gel electrophoresis purified from gel and joined in a ligation reaction. After transformation, *E. coli* cells were selected on an LB medium with Zeocin™. The resulting plasmids were propagated and extracted from transformants. pPICZ-PHO5 and pPICZ-eGFP plasmid structures were verified using PCR and restriction analysis (Appendix A).

The gene encoding recombinant Neo-2/15 protein was synthesized de novo (Evrogen, Moscow, Russia) and cloned into a pUC57 plasmid. The nucleotide sequence was designed to correspond to the Neo-2/15 amino acid sequence presented in [49] and was optimized for *K. phaffii* (Appendix A). *XhoI* and *XbaI* sites and technical sequences were added at the 3′ and 5′ ends of the sequence. pUC57-Neo and pPICZalpha B plasmids were cut by *XhoI* and *XbaI* restriction enzymes. pPICZalpha B vector was also dephosphorylated using the FastAP enzyme. After electrophoretic separation and gel extraction, vector (pPICZalpha backbone) and insert (Neo-2/15 sequence) fragments were joined in a ligation reaction. After transformation, *E. coli* cells were selected on an LB medium with Zeocin™. The resulting plasmids were propagated and extracted from transformants. The pPICZ-Neo plasmid structure was verified using PCR and restriction analysis (Appendix A).

To compare the methods for plasmid generation, pPIC9-Neo was constructed using standard procedure, and pPICK-PHO5 and pPICK-Neo were generated using PCR amplification and in vivo cloning of *E. coli* (iVEC).

To obtain pPIC9-Neo, pPICZ-Neo and pPIC9 were cut with *SacI* and *AgeI* restriction enzymes. pPIC9 was also dephosphorylated using the FastAP enzyme. After electrophoretic separation and gel extraction, vector (pPIC9 backbone) and insert (fragment with Neo-2/15 sequence) were joined in a ligation reaction. *E. coli* cells were selected on LB medium with ampicillin. Screening of transformants was performed using colony PCR with PAOX1-F and NeoRQ-R primers. Resulting plasmids were propagated and extracted from transformants. The pPIC9-Neo plasmid structure was verified using PCR and restriction analysis (Appendix A). 

For the iVEC procedure, pPICZ-PHO5 and pPICZ-Neo vector backbones without Zeocin™ resistance gene were amplified using iVEC-F and iVEC-R primers. The KanR gene from Tn5 encoding an aminoglycoside 3″-phosphotransferase (APH 3′ II) was amplified using a Kan-F/ Kan-R pair of primers and pFA6a-kanMX6 plasmid [50] as a template. After electrophoretic separation, the resulting fragments were extracted from agarose gel. In total, 10 µL of a water solution containing 0.1 pM of vector fragment and 0.3 pM of insert fragment (*KanR* gene) was used for bacterial transformation. Screening of transformants was performed using colony PCR with PAOX1-F/NeoRQ-R primers for pPICZ-Neo and PHO5-F/PHO5-R primers for pPICZ-PHO5. The resulting plasmids were propagated and extracted from transformants. pPICK-PHO5 and pPICK-Neo plasmid structures were verified using PCR and restriction analysis (Appendix A).

### 2.8. Cell Transformation

For bacterial transformation, calcium-treated competent *E. coli* cells were prepared [51]. For the TfB1 buffer, a solution of 15% glycerol, 10 mM CaCl_2_, 100 mM KCl, and 30 mM CH_3_COOK was prepared. pH was adjusted to 6.8 and then MnCl_2_*4H_2_O was added until a 10 mM concentration was reached. For the TfB2 buffer, a solution of 15% glycerol, 75 mM CaCl_2_, and 10 mM KCl was prepared. pH was adjusted to 6.8 and then MOPS was added until a 10 mM concentration was reached. *E. coli* DH5α was grown overnight in 10 mL of TYM medium. In total, 1 mL of this culture was inoculated into 100 mL of TYM to make the inoculum. The culture was grown in a shaking incubator at 30 °C until OD_550_ reached 0.35. Cells were centrifuged for 10 min at 5000 rpm (3000× *g*), 4 °C. The pellet was re-suspended in 50 mL of cooled TfB1 buffer and incubated in a water/ice bath for 40 min. Then, cells were centrifuged at 5000 rpm (3000× *g*) for 15 min. The resulting pellet was re-suspended in 2 mL of cooled TfB2 buffer. Cell suspension was divided into 200 µL samples and stored at −70 °C.

Competent cells were transformed with plasmid DNA (1 µL of approximately 100 ng/µL solution), with ligation mixes (10 µL), or with iVEC mixes (10 µL). DNA solutions were added to 90 µL of a competent *E. coli* cell suspension and incubated for 30 min at 0 °C in a water/ice bath. Following incubation, the mixtures were heated at 42 °C for 30 s for heat shock and the tubes were transferred into a water/ice bath for 15 s. After addition of 900 µL of LB, the samples were kept at 37 °C for 1 h with regular stirring by flipping the tubes. Then, samples were centrifuged for 2 min at 6000 rpm (2400× *g*). Pellets were resuspended in 50 µL of LB, plated on solid LB with an antibiotic and cultivated at 37 °C.

Yeast transformations were carried out using electroporation in strict accordance with the protocol [52]. In total, 2 μg of *SacI* linearized vectors was used for the direct transformation of *K. phaffii* cells with pPICZ-PHO5, pPICZ-eGFP, pPICZ-Neo, pPIC9-Neo, pPICK-PHO5 and pPICK-Neo. For *K. phaffii* transformation with PCR fragments, an equimolar mix with 0.5 pM of each fragment was used.

### 2.9. Reporter Gene Assays

Acid phosphatase (ACP) activity was determined qualitatively [53] and quantitatively [54]. For qualitative analysis, a solution of α-naphthyl phosphate and Fast Blue B (1 mg per 1 mL) in 0.1 M Na-citrate buffer (pH 4.5) was prepared. A paper filter was soaked in this solution and placed on the surface of plates with grown yeast colonies. After 10 min, the staining of the yeast colonies was assessed.

For quantitative analysis of ACP activity, 100 µL of yeast cell culture was added to 800 µL of 0.1 M Na-citrate buffer (pH 4.5). Then, 100 µL of substrate solution (0.15 M p-nitrophenyl phosphate) was added. The mixture was stirred and incubated for 20 min at 30 °C. In total, 500 µL of 1 M NaOH was added to stop the reaction. During the reaction, acid phosphatase catalyzes the cleavage of the phosphate group from p-nitrophenyl phosphate and its conversion to colored product p-nitrophenol. The specific activity of ACP was designated as the ratio of the optical density of the reaction mix at 410 nm to the density of cell suspension at 550 nm.

Synthesis of eGFP in *K. phaffii* cells was analyzed using in-house equipment directly on plates with yeast colonies. The plates were illuminated using a blue light-emitting diode (480 nm, XLite). Roscolux #12 Straw (Rosco, Stamford, CT, USA) was used as an emission filter.

## 3. Results

### 3.1. Use of iVEC for Rapid Change of Selectable Markers in pPICZ Series Vectors

pPICZ-PHO5 plasmid was obtained using common molecular cloning techniques. This plasmid serves as an example of vectors, generated for heterologous expression of genes of interest in *K. phaffii* cells. It contains the *S. cerevisiae PHO5* acid phosphatase (ACP) reporter gene under the control of the *AOX1* gene promoter. To provide new possibilities to use common vectors based on the pPICZ series, we demonstrate a method for rapid change of selectable markers.

The procedure comprised the following steps (Figure 2): (1) The insert fragment was PCR-amplified with Kan-F and Kan-R primers using the pFA6a-kanMX6 plasmid as a template. These primers were designed for PCR amplification of KanR gene which provides kanamycin and G418 resistance. (2) The vector fragment (backbone) was PCR-amplified with iVEC-F and iVEC-R primers using the pPICZ-PHO5 plasmid as a template. This fragment corresponds to all parts of the vector except the coding sequence of the Zeocin™ resistance gene. (3) Fragments were separated by agarose electrophoresis and extracted from gel. (4) Purified DNA fragments were directly used for *E. coli* transformation. Kan-F/Kan-R and iVEC-F/iVEC-R pairs of primers were designed to contain overlapping regions (Appendix A). This allows combining the fragments into pPICK-PHO5 plasmid during in vivo cloning of *E. coli* (iVEC). In this plasmid, the *KanR* gene is placed under the control of *EM7* and *TEF1* gene promoters. This allowed for (5) selecting transformants on an LB medium with kanamycin. (6) To ensure that resulting plasmids contained the expression cassette, PCR screening of bacterial transformants (colony PCR) was performed using PHO5-F and PHO5-R primers for the *PHO5* reporter gene. Six out of eight colonies demonstrated positive results. (7) The pPICK-PHO5 plasmid was extracted from one of the transformants and its structure was verified using PCR and restriction analysis (Appendix A).

pPICK-PHO5 and pPICZ-PHO5 (as a control) were linearized by *SacI* and used to transform the *K. phaffii* X-33 strain. The transformants were selected on YEPDS media containing G418 and Zeocin™, respectively. Thus, K1-ACP-X-33 (*P_AOX1_-PHO5 KanR*) and Z1-ACP-X-33 (*P_AOX1_-PHO5 ZeoR*) *K. phaffii* strains were obtained. Qualitative analysis of reporter acid phosphatase activity on the surface of yeast colonies revealed *PHO5* reporter gene activity in the strains (Figure 3a). Thus, our results demonstrate that the pPICK-PHO5 vector generated by rapid change of a selectable marker is suitable for efficient delivery of expression cassette in the *K. phaffii* genome.

To generate the *K. phaffii* strain with two expression cassettes, the Z1-ACP-X-33 strain was transformed with the pPICK-PHO5 vector linearized by *SacI*. Transformants were selected on YEPDS media with G418. Thus, the KZ2-ACP-X-33 (*2xP_AOX1_-PHO5 ZeoR KanR*) *K. phaffii* strain was obtained. Quantitative analysis of reporter acid phosphatase activity produced by KZ2-ACP-X-33 with two copies of the expression cassette demonstrated that the *PHO5* reporter gene was 1.8 times more active than in parental Z1-ACP-X-33 strain carrying one copy (Figure 3b). Thus, our results demonstrate that the presented procedure allows for generating pPICZ-based vectors with different selectable markers, which can be used for the integration of multiple expression cassettes into *the K. phaffii* genome.

### 3.2. Use of Rapid Change of Selectable Markers for Generation of K. phaffii Strains for Efficient Production of Recombinant Neo-2/15 Protein

Neo-2/15 [49] was used in this study as a model protein to demonstrate that the method of rapid change of selectable markers is useful for productive *K. phaffii* strain generation. The pPICZ-Neo plasmid was obtained, which contains the Neo-2/15 gene under the control of the *AOX1* gene promoter. Sequences corresponding to *S. cerevisiae* alphaMF signal, c-myc epitope and 6x-His tag were added in-frame with Neo-2/15 coding sequence to provide secretion, detection and purification of recombinant protein (Appendix A).

The pPIC9-Neo plasmid with the *HIS4* gene as a selectable marker was constructed using common molecular cloning techniques. This required restriction digestion of the pPIC9-Neo and pPICZ-Neo plasmids with two endonucleases, dephosphorylation of vector, agarose gel electrophoresis, extraction of DNA fragments from gel, ligation, bacterial transformation, selection of transformants and plasmid analysis (Appendix A). 

For comparison, the pPICK-Neo plasmid was generated using the described rapid change of the selectable marker procedure (Figure 2). This required PCR amplification of the vector and insert (pPICZ-Neo fragment and *KanR* gene), agarose gel electrophoresis, extraction of DNA fragments from gel, bacterial transformation, selection of transformants and plasmid analysis. This approach appeared to be less labor-intensive and time-consuming, particularly because of skipping the ligation step.

Structures of pPICZ-Neo, pPIC9-Neo and pPICK-Neo plasmids were confirmed using PCR and restriction analysis (Appendix A). The *K. phaffii* GS115 (*his4*) strain was successively transformed with these plasmids after linearization by *SacI*. First, it was transformed with a pPIC9-Neo vector. After selection in an MD medium without histidine, the N1-GS115 (*P_AOX1_-Neo2/15 HIS4*) strain carrying only one expression cassette was obtained. This strain was transformed using the pPICZ-Neo vector with the following selection on YEPDS with Zeocin™. The resulting N2-GS115 (*2xP_AOX1_-Neo2/15 HIS4 ZeoR*) contains two copies of the expression cassette. In turn, it was transformed using the pPICK-Neo vector with the following selection on YEPDS with G418. Thus, the N3-GS115 (*3xP_AOX1_-Neo2/15 HIS4 ZeoR KanR*) strain with three copies of the expression cassette was obtained. 

To demonstrate the synthesis and secretion of recombinant Neo-2/15 protein by *K. phaffii*, the N1-GS115 strain was cultivated in two stages. First, it was grown in a BMGY medium with glycerol. Then, cells were transferred into a BMMY medium with methanol for induction of the *AOX1* promoter. Synthesis of recombinant protein was detected using electrophoresis and Western blot hybridization (Figure 4a). 

We analyzed the efficiency of Neo-2/15 synthesis in relation to the number of expression cassettes in the genome of the producer strain. N1-GS115, N2-GS115, and N3-GS115 *K. phaffii* strains were cultivated using the same two-stage protocol. The amount of synthesized recombinant protein was evaluated using electrophoresis (Figure 4b). Further analysis of electrophoregrams demonstrated that the N2-GS115 strain carrying two copies of expression cassette synthesizes 1.9 times more Neo-2/15 than the N1-GS115 single-copy strain. At the same time, N3-GS115 with three copies synthesizes 2.3 times more Neo-2/15 than N1-GS115 (Appendix A). Because N3-GS115 demonstrated only 21% increase in protein synthesis in comparison to N2-GS115, we used real-time PCR to show that N3-GS115 contains three copies of the expression cassette (Appendix A).

Our findings demonstrate that the described procedure of rapid change of selectable markers is reliable for the generation of *K. phaffii* strains using multiple selectable genes. The method is comparable to common molecular cloning techniques but less time- and labor-consuming.

### 3.3. Use of PCR and Split-Marker-Based Approach for Generation of K. phaffii Mut^S^ Strains with Two Expression Cassettes

Along with the pPICZ-PHO5 vector that contains the *PHO5* acid phosphatase reporter gene, a similar pPICZ-eGFP plasmid was obtained. The plasmid contains the enhanced green fluorescent protein (eGFP) reporter gene under the control of the *AOX1* gene promoter. With these plasmids, we demonstrate a method for generation of Mut^S^ *K. phaffii* strains using pPICZ series vectors.

The procedure comprised the following steps (Figure 5): (1) The first fragment of the pPICZ-PHO5 vector was amplified using PAOX1-F and ZeoUp primers. It contains a *PHO5* gene expression cassette with one part of the *ZeoR*-selectable marker. (2) The second fragment of the pPICZ-eGFP vector was amplified using ZeoDown2 and 3′AOX1-R primers. It contains an *eGFP* gene expression cassette with another part of the *ZeoR*-selectable marker. (3) Fragments were extracted from the reaction mix. (4) An equimolar mix of two fragments (0.5 pM each) was used directly for the transformation of *the K. phaffii* X-33 strain. Single fragments were used to control the transformation. (5) Yeast transformants were selected on YEPDS medium with Zeocin™. (6) The resulting colonies were transferred to MM media with methanol to select Mut^S^ strains and analyze the activity of the reporter genes.

Transformation of the *K. phaffii* X-33 strain yielded 2253 ± 369 colonies (mean for three replicas ± SD) for an equimolar mix of two fragments (0.5 pM each). Control transformation with single fragments yielded 49 ± 26 colonies for the first fragment and 28 ± 4 colonies for the second fragment (Appendix A). Control transformation without DNA did not result in any Zeocin™ resistant colonies.

The growth of transformants on MM medium with methanol which corresponds to their phenotype (Mut^+^ or Mut^S^) was examined. Analysis of reporter gene activity was also performed using qualitative methods (Figure 6 and Appendix A). 

Mut^+^ colonies comprised 80.1% of all transformants resulting from recombination events that differ from the desired variant presented in Figure 5. Such transformants demonstrate different phenotypes regarding reporter gene activity. 

The desired Mut^S^ colonies represented only 19.9% of all transformants but were easily selected on the MM medium. All of the analyzed Mut^S^ transformants demonstrated the Pho5^+^/eGFP^+^ phenotype and carried both *PHO5* and *eGFP* reporter genes. Thus, our results demonstrate that PCR and split-marker-based approaches allow for generating and selecting the Mut^S^ *K. phaffii* strains with two expression cassettes integrated into the genome. 

### 3.4. Use of PCR and Split-Marker-Based Approach for Generation of K. phaffii Strains for Efficient Production of Recombinant Neo-2/15 Protein

Neo-2/15 [49] was also used in this study as a model protein to demonstrate that the described PCR and split-marker-based approaches (Figure 5) are useful for the generation of productive *K. phaffii* strains. The pPICZ-Neo plasmid served as a template in separate PCR reactions with PAOX1-F/ZeoUp and ZeoDown2/3′AOX1-R pairs of primers. The two resulting PCR fragments were separated by agarose electrophoresis and extracted from gel. An equimolar mix of these fragments (0.5 pM each) was directly used for *K. phaffii* X-33 strain transformation. Yeast transformants were selected on a YEPDS medium with Zeocin™. The resulting colonies were transferred to MM media with methanol for Mut^S^ strain selection.

Genomic DNA was extracted from Mut^S^ transformants and integrated expression cassettes were analyzed using real-time PCR (Appendix A). It was shown that among five transformants, four contained two copies of the Neo-2/15 gene as expected, while one contained three copies.

One Mut^S^ transformant with two copies of the expression cassette (named SN2-X33 (*aox1::PAOX1-Neo2/15-ZeoR-PAOX1-Neo2/15*)) was selected for analysis of Neo-2/15 protein synthesis. Along with the control Mut^+^ strain N1-GS115, it was cultivated first in a BMGY medium for biomass growth and then in a BMMY medium for methanol induction and recombinant protein synthesis. The results were analyzed using electrophoresis (Figure 7).

Analysis of electrophoregrams showed that the SN2-X33 Mut^S^ strain carrying two copies of the expression cassette synthesizes 4.3 times more Neo-2/15 than the N1-GS115 single-copy strain (Appendix A). These results demonstrate that PCR and split-marker-based approaches can be used for the generation of *K. phaffii* Mut^S^ strains that efficiently produce recombinant proteins.

## 4. Discussion

*K. phaffii* yeast is now known not only as an efficient production host in biotechnology but also as an emerging model organism and an instrument for routine small-scale synthesis of recombinant proteins. Among the various methods and materials available to work with *K. phaffii,* some are more prominent and well spread than others. For example, pPICZ series vectors are widely used due to their effectiveness and long presence on the market. However, all methods and techniques have their advantages and limitations. In this work, we describe two alternative methods for expanding pPICZ vector applications. These methods provide rapid change of selectable markers and direct *K. phaffii* Mut^S^ strain generation that have not been previously reported for these vectors.

Several markers based on antibiotic (e.g., Zeocin™ and G418) resistance genes can be used for the selection of *K. phaffii* transformants. Vectors with such resistance markers are available (e.g., pPICZα B or pKANα B [27]). They allow the generation of *K. phaffii* strains with multiple integrations using high concentrations of antibiotics. However, this approach is often associated with the problem of multicopy strain stability. Multiple vector copies that are integrated into the *K. phaffii* genome create repeat regions of homology. These regions can be involved in loop-out recombination, resulting in expression cassette removal and instability of multicopy strains during prolonged cultivation or storage [33]. 

For the two-copy clone generated using standard transformation with a pPICZ-derived vector, 12 recombination events are possible (Appendix A). Some recombination events result in the elimination of both vector copies, but others lead to the elimination of only one of the expression cassettes. The resulting cells are still Zeocin™-resistant and their identification and elimination pose a challenge. 

For the two-copy clone generated using vectors with different selectable markers, recombination events result in the loss of at least one of the selectable genes (Appendix A). Thus, phenotypic analysis and verifying the preservation of the selectable marker can help exclude the clones that lost the expression cassette due to recombination (Appendix A). Therefore, strains with multiple selectable markers coupled with expression cassettes may be considered more suitable for prolonged cultivations, storage and repetitive use. However, to generate such strains, multiple vectors and several subsequent transformations are required. The problem can be solved by in vivo cloning of *E. coli* (iVEC) [55,56].

iVEC-F and iVEC-R primers were designed to provide amplification of a fragment of pPICZ series vectors. This fragment corresponds to all parts of the vector except the Zeocin™ resistance gene coding sequence. Kan-F and Kan-R primers were designed for PCR amplification of the *KanR* gene that provides kanamycin and G418 resistance. Because only the coding sequence of this gene is amplified, not only *K. phaffii* vectors, but also different available plasmids with this gene, can be used as templates. In this study, we used the pFA6a-kanMX6 plasmid which was developed for work on fission yeast *Schizosaccharomyces pombe* [50]. In the iVEC procedure, the *KanR* coding sequence is integrated under the control of *EM7* and *TEF1* promoters instead of the Zeocin™ resistance gene. 

During iVEC, the combination of DNA fragments into circular plasmids inside *E. coli* does not seem to be the result of recombination but rather depends on exonuclease activity [57]. Therefore, if the recombinant gene coding sequence is used as an insert, the expression cassette structure should be thoroughly checked in each resulting plasmid. In this study, we present an alternative approach based on changing not the gene of interest, but the selectable marker. Thus, when selecting media with kanamycin, non-functional plasmids containing incorrect or damaged inserts are excluded. This allows us to efficiently obtain functional plasmids with different selectable markers from the existing pPICZ-based vectors. 

Advantages of the iVEC method include the rapid generation of ‘empty’ pPICZ vectors for later use. Also, it can be used for the integration of additional copies of the gene of interest to increase the efficiency of protein production in *K. phaffii*.

This approach can be a part of strategies for the efficient generation of novel *K. phaffii* producer strains. When producing a recombinant protein with *K. phaffii,* the key step is to demonstrate that it is synthesized and secreted even at low levels. Then, there are different methods to improve production with multiplication of expression cassettes being one of the most effective. Thus, for initial experiments on the production of recombinant protein, the pPICZ-based vector containing the gene of interest and corresponding strain can be generated using common techniques. Then, producer strain efficiency can be enhanced by transformation with plasmids containing different selectable markers. Also, already existing *K. phaffii* strains generated using pPICZ-based vectors can be relatively easily modified using the described approach.

Other methods to enhance *K. phaffii* productivity rely on co-expression of different genes along with the gene of interest. This includes genes encoding transcription factors [58], chaperones [59,60], enzymes [60,61], etc. Additional selectable markers are required to deliver these genes in the *K. phaffii* genome along with selectable markers used for the gene of interest. The approach described in this paper allows for rapid changing of the selectable marker either in plasmids with auxiliary genes or in plasmids with the gene of interest. It can be helpful for the generation of *K. phaffii* producer strains with the co-expression of auxiliary genes enhancing the synthesis of recombinant protein.

It should also be mentioned that different antibiotics for selection come at different prices. Some of them may also be less available in different laboratories and countries. A rapid replacement of selectable markers overcomes these limitations. It may be proposed that this technique will work not only for the Zeocin™ and G418 resistance genes but also for other antibiotic-based selectable markers.

Vectors that are used for *K. phaffii* have different advantages and limitations. The pPIC9 vector allows for the direct generation of Mut^S^ strains by replacement of the *AOX1* locus with expression cassettes [39], although it requires a corresponding auxotrophic strain (e.g., GS115). Integration of several copies of the gene of interest using pPIC9 is difficult because it implies the generation of cumbersome plasmids with repeating expression cassettes. pPICZ series vectors can be used to transform wild-type strains and allow for the selection of multiple integration events. However, they do not allow us to directly generate Mut^S^ transformants.

In this study, we present an approach to generate Mut^S^ strains using conventional pPICZ series vectors. This technique is based on PCR amplification of two fragments using the pPICZ plasmid with the gene of interest as a template. Each fragment contains an expression cassette and a part of the Zeocin™ resistance gene. These parts of the *ZeoR* gene overlap, but separately they do not provide antibiotic resistance. During the co-transformation of *K. phaffii* with both fragments, they are joined within the cell producing functional genes via homologous recombination. The second act of recombination occurs between the *AOX1* gene locus in the *K. phaffii* genome and the *AOX1* promoter in the first fragment. Another act of recombination occurs between the *AOX1* terminator in the second fragment and the corresponding part of the *AOX1* gene (Figure 5). All these events result in *AOX1* locus substitution with two expression cassettes flanking the *ZeoR* gene. Thus, the described approach allows for generating the *K. phaffii* Mut^S^ strains with two copies of expression cassettes using only pPICZ series vectors without the need for preliminary modification. Our method also provides higher strain stability in comparison with standard pPICZ transformation due to a lower number of homology regions (Appendix A). Even if homologous recombination does occur, the resulting construct with a single cassette is stable due to the absence of homologous regions within the *AOX1* locus. 

The use of overlapping selectable marker fragments for yeast transformation was described previously and is known as the “split marker” approach. It proved to be efficient for introducing deletions in *K. phaffii* and other yeast genomes [62]. In this paper, we present ZeoUp and ZeoDown2 primers that may be helpful in other applications where the *ZeoR* gene is used as a “split marker”.

It is necessary to mention that the desired Mut^S^ strains obtained in this study comprise only 19.9% percent of all transformants. However, this issue is resolved by the selection of transformants with the Mut^S^ phenotype on a medium with methanol. This is not an additional step as the same procedure is required when working with the pPIC9 vector [39].

A relatively low number of Mut^S^ transformants can be explained by the distinctive features of *K. phaffii* recombination. In contrast to *S. cerevisiae, K. phaffii* has a lower efficiency of homologous recombination (HR) and relies on non-homologous end joining (NHEJ) for double-strand break reparation. Thus, *K. phaffii* tends to integrate single or multiple linear DNA fragments into random genome loci, leading to a high number of undesirable transformants [63]. Deletion of NHEJ-related proteins such as Ku70 (telomeric Ku complex subunit) and Dnl4 (DNA ligase IV) was shown to be an effective tool for improving the HR-to-NHEJ ratio in *K. phaffii* and reducing improper integrations [63,64,65]. In further work, the number of desired Mut^S^ transformants obtained by the split marker approach may be increased by strain modifications like *KU70* and *DNL4* gene deletions. Also, the modification of the pPICZ vector, for example, the extending of vector regions involved in HR and integration (e.g., 3′*AOX1* TT), can help increase the number of transformants with desired integrations. 

In this study, we used the rapid marker change and split marker approaches to demonstrate the potential of the described techniques to generate multicopy *K. phaffii* strains producing interleukin-2 (IL-2) mimetic Neo-2/15 [49]. IL-2 is known for its significant antitumor activity with severe toxicity and various side effects. A wide spectrum of protein engineering approaches including computational de novo design has been used to improve IL-2 antitumor properties and reduce toxicity. Neo-2/15 is one of the de novo-designed cytokine mimetics. It was developed as an IL-2 receptor agonist with eliminated binding to alpha subunits and enhanced affinity to beta and gamma subunits of the IL-2/IL-15 receptors [49]. Neo-2/15 is considered a prospective therapeutic agent possessing substantial antitumor activity with reduced toxicity. It may also be used in cytokine cocktails in fundamental research. In this study, we demonstrate that *K. phaffii* provides an efficient synthesis of Neo-2/15 and can potentially be used to produce similar cytokine-like proteins.

The analysis of the synthesized protein in *K. phaffii* strains obtained by using multiple selectable markers has shown that a change in expression cassette copy number does not lead to a proportional change in Neo-2/15 synthesis. The N2-GS115 strain has shown almost a 2-fold change in Neo-2/15 synthesis in comparison to N1-GS115, while N3-GS115 was only 2.3 times more productive. However, real-time PCR analysis of the strain showed three copies of the target gene in the genome. In contrast, the SN2-X33 strain containing two Neo-2/15 copies has shown a 4-fold increase in protein synthesis in comparison to N1-GS115. These results may be explained by differences in the strain phenotypes regarding methanol utilization. The SN2-X33 strain obtained by split-marker-mediated transformation has a Mut^S^ phenotype, while the other strains have Mut^+^. Higher recombinant protein production by Mut^S^ strains can be explained by the reduced protein synthesis workload on the cell and low rates of methanol consumption. Several previous studies have shown that Mut^S^ strains also provide increased production of target proteins [37,38]. At the same time, the influence of the original strains (X-33 and GS115) used for SN2-X33 and N(1-3)-GS115 strain generation on protein synthesis cannot be excluded.

## 5. Conclusions

pPICZ series vectors are widely used for the production of recombinant proteins in *K. phaffii*. In this study, we present a PCR-based way for modification of these well-known vectors providing rapid change of selectable markers. This technique allows easy generation of plasmids with different selectable markers without the need to obtain new specific vectors. We propose that such an approach is beneficial for laboratories that do not constantly work with *K. phaffii* or use this organism for recombinant protein production for ongoing experiments. 

We also describe a split marker approach for *K. phaffii* transformation allowing the generation of Mut^S^ strains using pPICZ series plasmids, which are not originally intended for Mut^S^ strain generation. Furthermore, this approach provides simultaneous integration of two expression cassettes by in vivo assembly of two DNA fragments, making it a promising tool for generating multicopy strains. We suggest that the efficiency of this approach can be further improved by HR-to-NHEJ ratio modification in the used strain as well as pPICZ vector modification.

By using Neo-2/15 cytokine mimetics as a model protein, we showed that both approaches described above can be used for the generation of multicopy strains. To the best of our knowledge, this is the first report of Neo-2/15 secretory expression in yeast, demonstrating that *K. phaffii* may potentially be applied for the production of similar cytokine-like proteins.

## Figures and Tables

**Figure 1 microorganisms-11-02297-f001:**
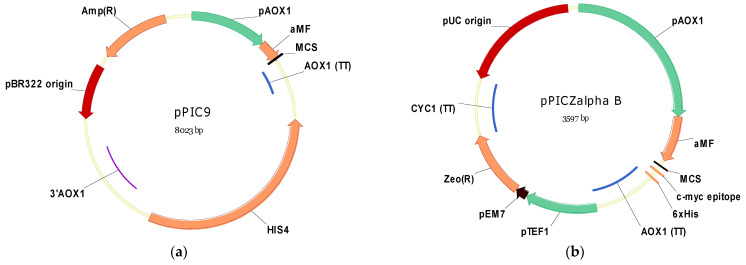
pPIC9 and pPICZalpha-B plasmid maps. (**a**) pPIC9 contains *AOX1* promoter sequence (pAOX1), alpha-factor sequence (aMF), multiple cloning site (MCS), *AOX1* transcription terminator (AOX1 (TT)), auxotrophic marker *HIS4, AOX1* 3′ fragment (3′AOX1), ampicillin resistance gene (Amp(R)) and origin of replication pBR322. (**b**) pPICZalpha-B contains *AOX1* promoter sequence (pAOX1), alpha-factor sequence (aMF), multiple cloning site (MCS), c-myc epitope and 6xhis tag sequence (6xHis), *AOX1* transcription terminator (AOX1 (TT)), Zeocin™ resistance gene (Zeo(R)) regulated by prokaryotic (pEM7) and eukaryotic (pTEF1) promoters, *CYC1* transcription terminator (CYC1 (TT)), and pUC origin of replication. Green arrows—eukaryotic promoters; brown arrows—prokaryotic promoter; orange arrows and lines—coding sequences; blue lines—transcription terminator sequences; red arrows—origin of replication; and purple line—*AOX1* 3′-fragment for homologous recombination.

**Figure 2 microorganisms-11-02297-f002:**
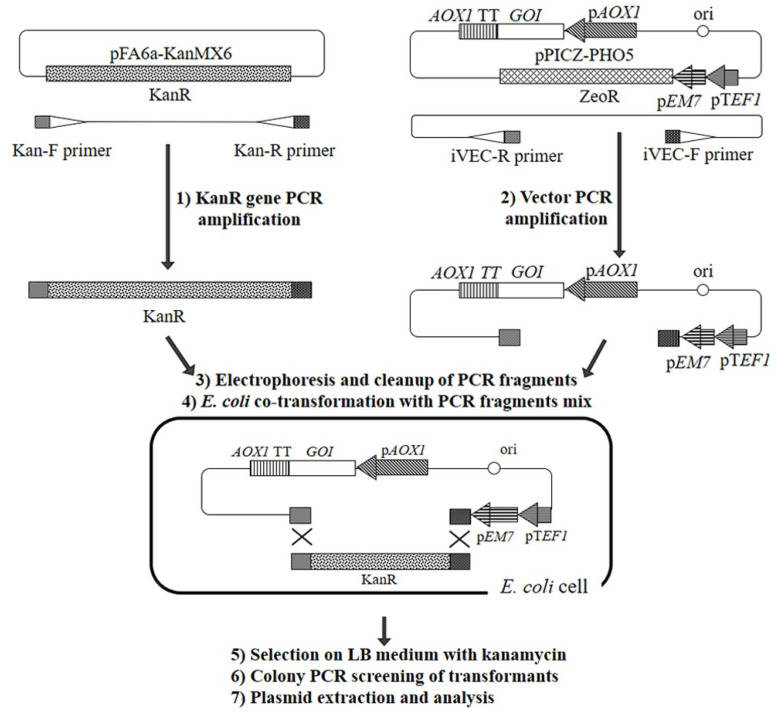
Overview of the procedure for rapid change of selectable marker in pPICZ series vectors. *pAOX1*—*AOX1* promoter; *AOX1* TT—*AOX1* transcription terminator; *GOI*—gene of interest; p*EM7*—prokaryotic *EM7* promoter; p*TEF1*—eukaryotic *TEF1* promoter; ZeoR—Zeocin™ resistance gene; KanR—G418/kanamycin resistance gene.

**Figure 3 microorganisms-11-02297-f003:**
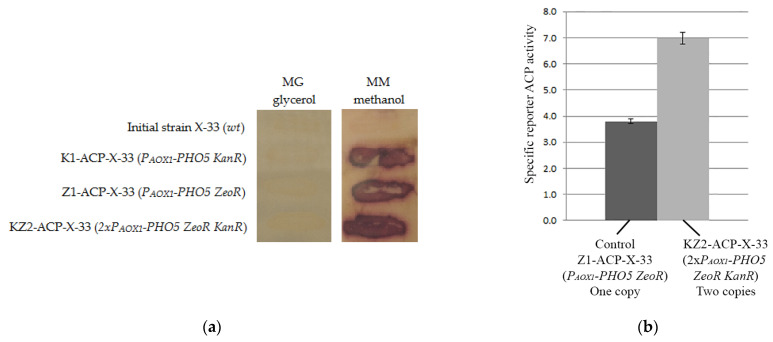
(**a**) Results of qualitative analysis of reporter ACP activity on the surface of the colonies of *K. phaffii* strains carrying expression cassettes with the reporter *PHO5* gene under the control of *AOX1* promoter. *AOX1* promoter is repressed on MG medium with glycerol. Its activation on MM medium with methanol leads to production of ACP. *K. phaffii* X-33 strain was used as a control. (**b**) Results of quantitative analysis of reporter ACP activity in cultures of Z1-ACP-X-33 strain carrying one expression cassette with *PHO5* gene and KZ2-ACP-X-33 strain with to copies of reporter construction. Mean specific activity of ACP (±SD) for measured four separate cultivations is presented.

**Figure 4 microorganisms-11-02297-f004:**
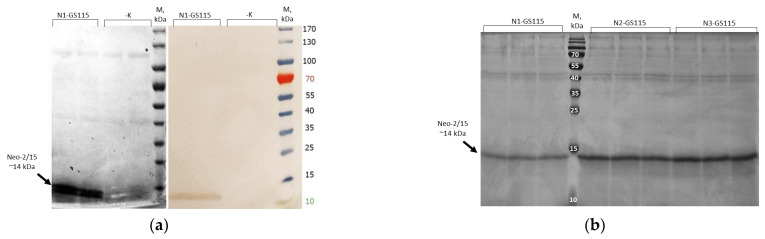
(**a**) Electrophoregram and western-blot analysis of proteins secreted by *K. phaffii* N1-GS115 strain (Novex™ Tris-Glycine Gels (gradient 8–16%) were used); (**b**) Electrophoregram of proteins secreted by *K. phaffii* N1-GS115, N2-GS115, N3-GS115 strains carrying from 1 to 3 Neo-2/15 expression cassettes (standard 6-15% PAAG was used).

**Figure 5 microorganisms-11-02297-f005:**
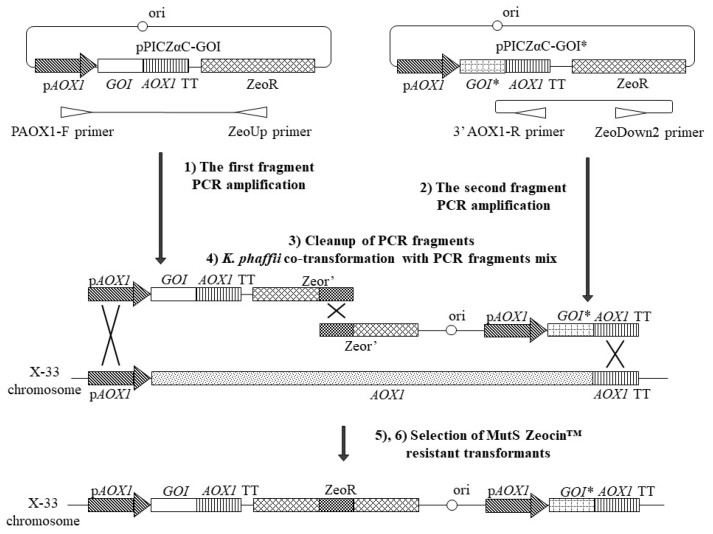
Overview of PCR and split-marker-based approach for generation of *K. phaffii* Mut^S^ strains with two expression cassettes. *AOX1*—alcohol oxidase 1 gene; *AOX1* TT—*AOX1* transcription terminator; *GOI—*gene of interest*; ori—replication origin; p*AOX1—AOX1* promoter; ZeoR—Zeocin™ resistance gene; Zeor’—truncated Zeocin™ resistance gene. **GOIs* can be either the same or different, so one or two different plasmids can be used as template for PCR amplification steps (1) and (2).

**Figure 6 microorganisms-11-02297-f006:**
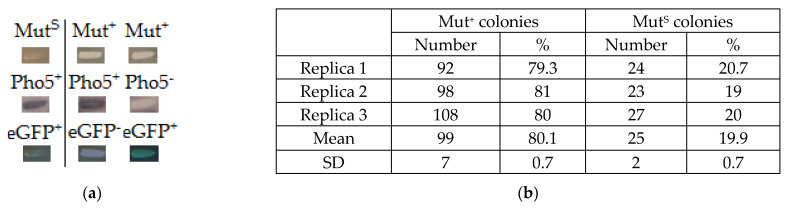
(**a**) Examples demonstrating different phenotypes of transformants obtained by PCR and split-marker-based approach. For Mut^+^ colonies, all possible phenotype combinations regarding reporter gene activity were observed (Pho5^+^/eGFP^+^, Pho5^+^/eGFP^−^, Pho5^−^/eGFP^+^, Pho5^−^/eGFP^−^). For Mut^S^ colonies, only the desired Pho5^+^/eGFP^+^ phenotype was observed. (**b**) Results of phenotype analysis. At least 90 colonies were analyzed for each transformation. Mean numbers of colonies were rounded to nearest integers.

**Figure 7 microorganisms-11-02297-f007:**
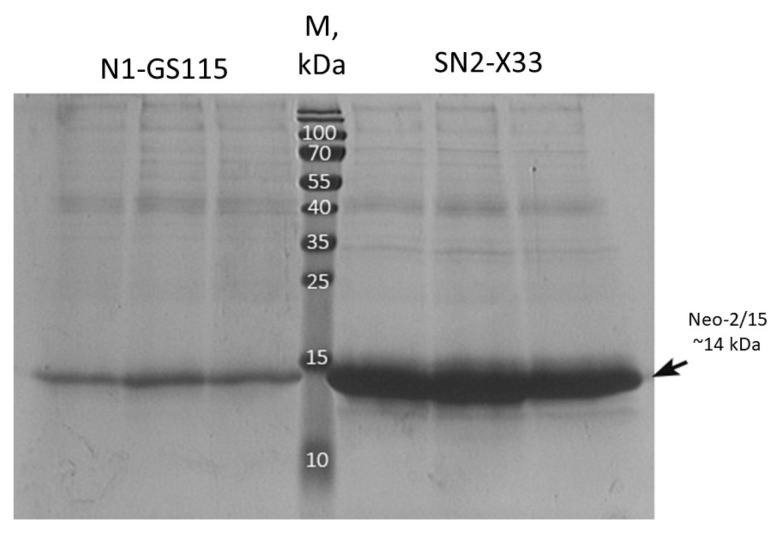
Electrophoregram of secreted proteins of *K. phaffii* strains N1-GS115 and SN2-X33.

**Table 1 microorganisms-11-02297-t001:** *K. phaffii* strains used in this study.

Strain	Genotype	Sourse
GS115	*his4*	Thermo Fisher Scientific, Waltham, MA USA
X-33	*wt*	Thermo Fisher Scientific, Waltham, MA USA
K1-ACP-X-33	*P_AOX1_-PHO5 KanR*	This study
Z1-ACP-X-33	*P_AOX1_-PHO5 ZeoR*	This study
KZ2-ACP-X-33	*2×P_AOX1_-PHO5 ZeoR KanR*	This study
S-ACP-GFP-X-33	*aox1*::*P_AOX1_-PHO5-ZeoR-P_AOX1_-GFP*	This study
N1-GS115	*P_AOX1_-Neo2/15 HIS4*	This study
N2-GS115	*2×P_AOX1_-Neo2/15 HIS4 ZeoR*	This study
N3-GS115	*3×P_AOX1_-Neo2/15 HIS4 ZeoR KanR*	This study
SN2-X33	*aox1*::*P_AOX1_-Neo2/15-ZeoR-P_AOX1_-Neo2/15*	This study

**Table 2 microorganisms-11-02297-t002:** Primers used in this study.

Name	Sequence (from 5′ to 3′ End)
NeoRQ-F	TCCAGCTGAAGAGAAGTTGGA
NeoRQ-R	CGGAGAAAATCCAGCTTTGA
ACT1-F	AGTGTTCCCATCGGTCGTAG
ACT1-R	GGTTCATTGGAGCCTCAGTC
PHO5-F	CGGGATCCCGAGATTACCAA
PHO5-R	CGGAATTCCAAAACTATTGT
eGFP-F	ATTACAGGATCCATGGTGAGCAAGGGCG
eGFP-R	ATTACAGAATTCTTACTTGTACAGCTCGTCCATGC
PAOX1-F	AACATCCAAAGACGAAAGG
3′AOX1-R	CACAAACGAAGGTCTCACTTA
ZeoUp	AGTTGACCAGTGCCGTTC
ZeoDown2	CGGAAGTTCGTGGACAC
iVEC-F	TCGATGAGTTTTTCTAAGGACTGACACGTCCGAC
iVEC-R	TTTTCCTTACCCATGGTTTAGTTCCTCACCTTGTC
Kan-F	GAGGAACTAAACCATGGGTAAGGAAAAGACTCAC
Kan-R	CGTGTCAGTCCTTAGAAAAACTCATCGAGCATC

## Data Availability

The data presented in this study are available in Appendix A.

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
