# Peer review of "Alternative PCR-Based Approaches for Generation of Komagataella phaffii Strains"

_microorganisms, 2023, doi:10.3390/microorganisms11092297_

Round 1

Reviewer 1 Report

The manuscript focuses on the construction of an expandable vector system for the expression of recombinant genes/proteins in K. phaffii. The antibiotic resistant KanR and ZeoR as well as His4 were used for selection. Expression of different target proteins was confirmed by Western blots and activity assays. For inducible expression, the methanol inducible promoter AOX1 was used. For efficient inducible expression, the corresponding genomic section of AOX1 was modified by homologous recombination to generate MutS strains. For this purpose, the pPICZ was modified to enable specific integration at the AOX1 locus.

Although the experimental procedure is presented logically, the text should be more compact and focused. In addition, there are some spelling mistakes and unclear formulations in the text: Lines 22-23 comma is missing. Lines 103-104 the sentence is unclear. Lines 110-112 comma missing. Lines 272-273 comma missing. 457-458, comma missing. Please use pM instead of picoM.

The illustrations of the plasmids in Figure 1 which were probably created with VectorNTI differ from the other illustrations in the manuscript and the Supporting information (S1,S3 and S5). These should be consistent. The colors of the feature map are not explained.

Both the cloning technology iVEC and the basic strategies for integration for protein expression are not new. Although the reviewer considers the creation of new plasmids for K. phaffii expression to be positive, the scientific depth is limited.

It is not clear from the manuscript why expression cassettes are not generated using conventional techniques such as restriction-based cloning. There is a wide range of different approaches from BioBrick to Gibbson. What is the advantage over these techniques?

The English should be improved.

Reviewer 2 Report

This is a manuscript purporting to show the development of novel methods for quick replacement of selection marker genes in K. phaffii vectors, and for the generation of MutS strains concomitant with the insertion of double copy of the transgene using a split-marker strategy.

The paper is scientifically sound. The results are clear. Statistical testing could be better (don't use SEMs as they only indicate how accurately we know the mean, not what the scatter of the data is; use SDs or CI95s instead).

However, I'm afraid there's no novelty to this manuscript at all. In vivo cloning using E. coli and split-marker strategies in yeast are well-established, old techniques. You've merely used them to delete AOX1 and replace markers. It's clever, but trivial.

And I'm not sure these are even the best ways to accomplish what you seek. Instead of iVEC, you could've easily generated linear fragments containing the new resistance gene using overlap PCR, thus sparing the need to go through the bacterium. I cannot see the speed-up you've suggest as being significant.

The reason people don't resort to these methods more often are manifold. Groups can simply use Gibson assembly to make their cassettes, strains that are already MutS are widely available. With marker loop-out recycling strategies such as the rhamnose method and Cre-loxP vectors (both described for K. phaffii), the need for multiple markers is diminished. There's also the issue that your marker-replacement method targets the construct always to the same region in the DNA, so iterative transformations will rely on ectopic insertions more and more.

I would also add that your split-marker strategy yields inserts with lots of repetitive regions that could loop out. I wonder how stable your transformants are at serial passaging.

All in all, I fail to see that what this manuscript adds much of significance to the list of genetic engineering strategies in K. phaffii. It seems to me that you ought to have focused on the bioprocess of the IL-2 mimetic you've first produced in this yeast.

The manuscript is legible, but the English is very stilted: most the articles, definite and indefinite, are missing. It needs a thorough correction from someone proficient in the language.

Round 2

Reviewer 2 Report

I've been persuaded that the tools presented in this paper can be of help for groups facing budgetary constraints or a dearth of available vectors. Your presentation now puts due emphasis on this fact.

Your explanation of the loop-out risk of the resulting constructs is detailed and serves as a "buyer beware" statement for potential users of this strategy. I would insist, though, that you still don't stress what the main issue is.

Yes, applying selective pressure with the antifungal drug helps ensure only cells that didn't lose the selectable marker will grow. It's good practice to always plate a strain fresh out of the freezer first onto selective agar.

However, the problem is getting these strains to grow in industrial bioreactors. The cost of adding antifungals to large culture vats is prohibitive, so strains must be stable enough to keep their genotype while growing in liquid medium without selective pressure. In this sense, the strategies analysed in figures S16 and S17 are very prone to loop-out recombination and loss of the GOI in the course of industrial production.

I'd suggest that you emphasise that the strategy in figure S18, while prone to lose the selectable marker, seems to generate constructs stable relative to at least one copy of the GOI. As you show, all three possibilities of the recombination lose the marker, but retain one GOI.

You also mention in your response letter that you have performed stability experiments. I strongly suggest that you include these results at the very least as a supplement, detailing how you performed the passaging and prolonged cultivations you mention. The fact that strains with only two copies are stable is encouraging: tell your readers that!

The quality of the manuscript has improved significantly, but some of the added paragraphs need a little more tweaking of the English. For example, "rapid change of selectable marker approach" is not an acceptable syntagma in English, a language that doesn't use chains of nouns like this. "A rapid replacement of selectable marker overcomes (…)" is much better. The use of the verb "to allow" also needs some revising (it's "allows for excluding" in lines 654-5, for example). But overall, it reads much better.

Author Response

The authors are sincerely grateful for the comments on the manuscript.

The English in the previously added paragraphs has been improved according to the comments. We also added the sentence that emphasizes the stability of strains generated using the split-marker approach. All changes can be viewed in review mode in Word document.

K. phaffii is indeed a popular microbial host for industrial-scale recombinant protein production. Additionally, the yeast is also widely used now in fundamental research to synthesize recombinant proteins to use in further experiments. Due to the different scales of production, the requirements for the producer strains, as well as the efforts for their generation, may differ. The industrial strains have to be highly productive and stable. Thus, it is reasonable to use expensive but efficient techniques for their generation, and carefully assess the risk of copy loss in multicopied strains. The main purpose of our work was to expand the set of tools to generate K. phaffii strains, to be used by laboratories for small-scale protein production. Therefore, we proposed the simple techniques to expand the applications of the widely spread pPICZ-based vectors.

In our work with Neo-2/15 protein, we obtained a large collection of strains with different numbers of Neo-2/15 expression cassettes and different phenotypes (MutS/Mut+), including those listed in the main text of the article. We also performed stability tests on the strains and presented the results of these experiments in Supplementary 7.